# The Distinctive Permutated Domain Structure of Periplasmic α-Amylase (MalS) from Glycoside Hydrolase Family 13 Subfamily 19

**DOI:** 10.3390/molecules28103972

**Published:** 2023-05-09

**Authors:** Yan An, Phuong Lan Tran, Min-Jee Yoo, Hyung-Nam Song, Kwang-Hyun Park, Tae-Jip Kim, Jong-Tae Park, Eui-Jeon Woo

**Affiliations:** 1Disease Target Structure Research Center, Korea Research Institute of Bioscience and Biotechnology, Daejeon 34141, Republic of Korea; 2Division of Animal, Horticultural and Food Sciences, Graduate School of Chungbuk National University, Cheongju 28644, Republic of Korea; 3Department of Food Science and Technology, Chungnam National University, 99 Daehak-ro, Yuseong-gu, Daejeon 34134, Republic of Korea; 4Department of Food Technology, An Giang University, Long Xuyen 880000, Vietnam; 5Vietnam National University-Ho Chi Minh City, Ho Chi Minh 700000, Vietnam

**Keywords:** MalS, domain permutation, CBM69, maltohexaose, glycoside hydrolase 13

## Abstract

Periplasmic α-amylase MalS (EC. 3.2.1.1), which belongs to glycoside hydrolase (GH) family 13 subfamily 19, is an integral component of the maltose utilization pathway in *Escherichia coli* K12 and used among Ecnterobacteriaceae for the effective utilization of maltodextrin. We present the crystal structure of MalS from *E. coli* and reveal that it has unique structural features of circularly permutated domains and a possible CBM69. The conventional C-domain of amylase consists of amino acids 120–180 (N-terminal) and 646–676 (C-terminal) in MalS, and the whole domain architecture shows the complete circular permutation of C-A-B-A-C in domain order. Regarding substrate interaction, the enzyme has a 6-glucosyl unit pocket binding it to the non-reducing end of the cleavage site. Our study found that residues D385 and F367 play important roles in the preference of MalS for maltohexaose as an initial product. At the active site of MalS, β-CD binds more weakly than the linear substrate, possibly due to the positioning of A402. MalS has two Ca^2+^ binding sites that contribute significantly to the thermostability of the enzyme. Intriguingly, the study found that MalS exhibits a high binding affinity for polysaccharides such as glycogen and amylopectin. The N domain, of which the electron density map was not observed, was predicted to be CBM69 by AlphaFold2 and might have a binding site for the polysaccharides. Structural analysis of MalS provides new insight into the structure–evolution relationship in GH13 subfamily 19 enzymes and a molecular basis for understanding the details of catalytic function and substrate binding of MalS.

## 1. Introduction

MalS is an integral component of the maltose utilization pathway in *Escherichia coli*, crucial to the effective utilization of maltodextrin. MalS is a gene that encodes periplasmic α-amylase. It is regulated by the MalT protein [1,2], which controls the synthesis of proteins responsible for the utilization of maltodextrin, such as MalE~K [1,3,4]. Maltodextrins enter the periplasm through specific maltoporins (LamB porins) in the outer membrane [5,6,7,8,9], where they are recognized by maltose-binding protein (MBP or MalE protein) as a substrate, and transport is facilitated via the transport system [6,10,11]. MalF, MalG, and MalK form binding complexes to transport maltooligosaccharides across the cytoplasmic membrane. These complexes are responsible for transporting maltooligosaccharides with limited glucose units [12,13,14,15]. MalS degrades linear dextrins by recognizing the substrate at the non-reducing end and preferentially liberating maltohexaose [9]. This product can then be transported across the cytoplasmic membrane via maltose ABC transporters.

α-Amylase is a commonly occurring glycoside hydrolase (GH). It cleaves α-1,4-glycosidic bonds in carbohydrates and oligosaccharides. It plays a significant role in starch digestion in humans, plants, and microorganisms and has various industrial applications, such as starch liquefaction and saccharification [16,17,18]. MalS is an α-amylase that breaks down maltodextrins in the periplasm, hydrolyzing the 1,4-glucosidic bond of amylose and long maltodextrin to release maltohexaose. MalS is an endohydrolase capable of cleaving at least three units of maltodextrin and releasing products such as glucose, maltose, maltotriose, as well as maltohexaose. MalS is unable to degrade α-CD, but it can degrade large cyclodextrins such as β-CD and γ-CD [1,9]. 

MalS is a member of glycoside hydrolase family 13 (GH13), the major family of glycoside hydrolases that include hydrolases, transglycosidases, and isomerases [19]. In the sequence-based classification system of carbohydrate-active enzymes (CAZy database, http://www.cazy.org, accessed on 12 April 2023) [20], α-amylase is one of the most commonly occurring glycoside hydrolases (GHs). Family GH13 is generally known as the main α-amylase family. The α-amylases classified in family GH13 share 4–7 conserved sequence regions (CSRs), catalytic machinery, and adopt the (α/β)8-barrel fold of the catalytic domain [21]. Within GH13, several subfamilies exhibit a higher degree of sequence similarity to each other than to members of other GH13 subfamilies, and α-amylase specificity is present in these subfamilies. Currently, 46 subfamilies of GH13 have been defined [22]. According to the carbohydrate-active enzymes (CAZy) database, MalS is classified as a member of GH13_19. Generally, GH13 enzymes have three domains in common: domain A contains the catalytic moiety, which is a catalytic domain with a typical (β/α)8- or TIM-barrel comprising eight parallel β-sheets surrounded by eight α-helices; domain B connects the third α-helix and third β-sheet of domain A, which is essential for substrate binding and has a variable length and structure; and domain C forms a β-sandwich structure designated as a putative saccharide-binding site [23]. 

The activity of MalS relies on divalent cations, particularly Ca^2+^, to enhance its performance [24]. Calcium ions are vital for the structure, function, and stability of α-amylases, especially for thermophilic varieties of a-amylase [25,26], as they maintain the correct protein conformation and prevent enzyme thermal inactivation [27]. However, because the catalytic site is distant from the calcium binding site, the calcium ions’ primary role in α-amylases is structural [28,29]. Most α-amylases are dependent on Ca^2+^, except for some Ca^2+^-independent α-amylases [30,31], whereas other α-amylases are inhibited by Ca^2+^ ions [32]. 

MalS is comprised of 676 amino acids with an additional region in the N-terminal, making it longer than typical extracellularly secreted α-amylases. This becomes evident when MalS is compared with known α-amylases. Sequence alignment was performed between MalS (UniProt: P25718) and typical α-amylases such as AMY1A_HUMAN (*Homo sapiens*, Human, UniProt: P0DUB6), AMYA1_ASPOR (from *Aspergillus oryzae* strain ATCC 42149/RIB 40, UniProt: P0C1B3), AMY_BACAM (*Bacillus amyloliquefaciens,* UniProt: P00692), AMY2_ECOLI (Cytoplasmic α-amylase, *E. coli* K12, UniProt: P26612), and AMY1_HORVU (*Hordeum vulgare,* UniProt: P00693); the conserved sequence regions (CSRs) (I-VII) are shown in red squares in Appendix A [33]. The sequence alignment showed that compared to typical α-amylases, MalS has more than 100 additional amino acids at the N-terminal. Although previous studies of MalS demonstrated that the enzyme exhibits catalytic properties similar to those of typical extracellular α-amylases, this domain structure suggests that MalS has a distinct feature. 

This study presents the crystal structure of MalS at a resolution of 2.7 Å, providing the first elaborated structure of MalS. Additionally, this is also the first crystal structure analysis for GH13 subfamily 19. In this study, the specific architecture of MalS was analyzed, and the substrate binding sites and calcium ion binding of the enzyme were studied. This study provides a molecular basis for understanding the catalytic characteristics of the important periplasmic α-amylase, which is commonly utilized among Enterobacteriaceae. 

## 2. Results and Discussion

### 2.1. Overall Structural Characteristics of MalS

The crystal structure of MalS, a periplasmic α-amylase from *E.coli* strain K12, was determined at a resolution of 2.7 Å. MalS (PDB: 8IM8, belongs to GH13_19) was crystallized in space group H:3, and structural analysis was performed using molecular replacement with CDI5 (cyclomaltodextrinase, PDB ID: 1EA9, belongs to GH13_20) as a template. Most of the polypeptide chains had well-defined electron densities, except for residues 1–119, which were not visible in the electron map. This suggests a greater degree of flexibility in these regions. MalS consists of four domains: domain A (residues 181–314, 436–645), domain B (residues 315–435), domain C (residues 120–180, 646–676), and the non-visible domain N (residues 1–119) (Figure 1A). Domain A contains the characteristic (β/α)_8_ structure (TIM barrel fold) found in many members of the α-amylase family and is known as the catalytic domain. Domain B connects the third α-helix and third β-strand of domain A and consists of a long loop and α-helix. Domain C (residues 120–180, 646–676) (shown in green and cyan in Figure 1B) consists of eight β-strands forming a β-sandwich. Domain C has a structure similar to that of the C domains of neopullulanase 1 (PDB ID: 1UH3, from *Thermoactinomyces vulgaris*, belongs to GH13_21) and cyclomaltodextrinase (PDB ID: 1EA9, belongs to GH13_20), with individual RMSD values of 0.743 Å (14 Cα atoms) and 2.926 Å (35 Cα atoms) (Figure 1B). Therefore, the evolutionary path of the MalS gene could be different from those of other GH13 subfamily 19 amylases. The sequence homology of MalS with those enzymes was very low. The MalS structure has clear disulfide bridges between C121 and C537, located in domains C and A (Figure 1C). Since domains C and A are connected, the disulfide bond serves to stabilize the structure and improve the thermostability of the protein.

MalS showed structural similarities to the neopullulanase 1 (PDB ID: 1UH3) that belongs to GH13_21 and the cyclomaltodextrinase (PDB ID: 1EA9) that belongs to GH13_20, with RMSD values of 3.69 Å (293 Cα atoms) and 1.99 Å (258 Cα atoms), respectively. However, MalS has a larger B domain in comparison to cyclomaltodextrinase and neopullulanase 1. The analysis of the conserved sequence regions V (CSR-V) in domain B of MalS showed PDIK in the CSR-V region, whereas cyclomaltodextrinase and neopullulanase have a PKLN motif in the CSR-V region (Appendix A) [34,35]. Since this motif is involved in calcium ion binding, this is one of the structural features of MalS. The calcium ion binding of MalS is further discussed in Section 2.3.

Intriguingly, domain C in MalS is formed by the combination of N-terminal residues 120–180 and C-terminal residues 646–676 (Appendix A). To our knowledge, this is the first C-domain structure in GH13 subfamily 19 that consists of two fragmented regions. In GH70, glucansucrases (EC 2.4.1.5) and 4,6-α-glucanotransferases (2.4.1.-) from lactic acid bacteria have similar permutated domain structures. Among them, GTF180-ΔN (glucansucrase from *Lactobacillus reuteri* 180, PDB: 3KLK) folds into five distinct linearly arranged domains, designated as A, B, C, IV, and V, but domains A to IV are organized by the discontinuous parts of the polypeptide chain. In this enzyme, only the C domain consists of one continuous stretch of amino acids (Appendix A). However, the domain permutation including C domain has not been reported in other GH13 enzymes.

To date, GH13 subfamily 19 consists mainly of *E. coli* MalS type enzymes. MalS shares a high degree of sequence similarity with the maltohexaose-producing amylase from Klebsiella pneumoniae (UniProt: Q9RHR1), which also belongs to the GH13_19 family. Interestingly, the AlphaFold-predicted structure of this enzyme has a unique feature of the C domain being composed of discontinuous parts of the polypeptide chain. Moreover, comparing the sequences and AlphaFold-predicted structures of periplasmic α-amylases from different genera, such as *Shigella flexneri* (UniProt: A0A0H2V321), *Salmonella typhimurium* strain LT2/SGSC1412/ATCC 700720 (UniProt: Q8ZL87), and *Citrobacter rodentium* strain ICC168 (UniProt: D2TJT7), showed high similarity with MalS. This suggests that MalS possibly originated from a common ancestor and has been conserved over a long evolutionary period. In other words, since MalS plays an important role in the survival of various bacteria, its conservation across diverse bacterial types further supports its ancient origin and evolutionary significance.

### 2.2. Binding Affinity and Structural Characterization of N Domain Predicted by AlphaFold2

Intriguingly, MalS has a strong ability to bind to oligosaccharides and polysaccharides. It has been discovered that MalS has high affinity for amylopectin and glycogen. To confirm MalS’s ability to bind to these substrates, the enzyme was loaded onto an amylose resin and then eluted using Tris-HCl buffer containing maltose, glycogen, or amylopectin. The SDS-PAGE results showed that MalS was eluted from the amylose resin by amylopectin and glycogen. When only Tris-HCl buffer was used as the eluent, MalS was still bound to the resin, and the enzyme was not detected in the elution (Figure 2).

This suggests that MalS probably contains a carbohydrate-binding module (CBM). However, the CBM domain was not found in the observed structure of MalS. Therefore, it was expected that the non-visible N domain might be the CBM domain. To confirm this, we predicted the structure of the N domain using AlphaFold2, which provided the architecture of the carbohydrate-binding domain. The Dali server indicated its association with a CBM69 domain, with an RMSD of 5.4 Å (71 Cα atoms) (PDB ID: 5X5S). Interestingly, the predicted structure of MalS by AlphaFold2 showed a pLDDT (per-residue local distance difference test) score of 92.8, pTM (per-template model quality) score of 0.879, and tol (tolerance) score of 0.337, whereas the N domain was connected by a loop in which the pLDDT score of <50 was significantly low (Appendix A). Based on this analysis, the N domain is expected to be highly flexible, explaining its lack of visibility in the crystal structure. In the predicted structure, the N domain was located close to the active site, and one of its loops formed a cover lid (Figure 3B). Future mutational studies of MalS will be needed to confirm the role of the N domain.

The CBM69 that has been characterized so far is the one found in AmyP [20,36]. The structure of recombinant CBM69, which was produced independently, was elucidated using NMR techniques [37]. This CBM69 partially unfolds in its free form, resembling a compact molten globule. After binding to β-CD, however, the module folds into a relatively rigid state [38]. The N domain of MalS could have a flexible structure similar to that of AmyP. Further study including co-crystallization with β-CD or other sugar ligands may help to elucidate the structure and function of the N domain.

In the first report of CBM69, a close evolutionary relationship between CBM69 and families CBM20 and CBM48 was suggested, as these families were previously shown to share a common ancestor. The binding sites in CBM69 have been hypothesized to potentially participate in α-glucan binding, based on similarities with CBM20 and CBM48 [36,39]. However, the three-dimensional structure of CBM69 has raised questions about its close relationship with CBM20 and CBM48. The fold topology and orientation of respective β-strands in CBM69 are different from those observed in CBM20 and CBM48 [38,40]. Similarly, the N domain of MalS also differs from typical CBM20 and CBM48 in the orientation of β-strands (Appendix A).

### 2.3. Essentials of Calcium Ion Binding to MalS

MalS is ion-dependent and its catalytic activity requires 2+ ions. Similar to typical α-amylases, MalS is predicted to be dependent on Ca^2+^. In 1997, Spiess demonstrated that MalS is active only in the presence of 2+ ions. Among many ions, Ca^2+^ was found to be the cation with which MalS showed the highest activity [24]. Based on high electron densities and coordination geometry, we identified two calcium ion positions in the MalS structure (Figure 4C, D). These Ca^2+^ ions were designated Ca1 and Ca2. The positions of these ions are conserved in many GH13 α-amylases and resemble those in the structures of *Anoxybacillus ayderensis* α-amylase (PDB ID: 5A2A, belongs to GH13_45), *Geobacillus stearothermophilus* α-amylase (PDB ID: 1QHO, belongs to GH13_2), and *Niallia circulans* cyclomaltodextrin glucanotransferase (PDB ID: 1EO5, belongs to GH13_2) (Appendix A). Ca2 is located in domain A, while Ca1 is located between domains A and B. Ca2 is situated at the back of the active center, whereas Ca1 is located near the -3, -4 subsite binding loop (Figure 4E).

Peptide residues that modulate Ca^2+^ commonly include Asp, Asn, and Glu. Ca^2+^ typically forms octahedral coordination geometries with coordination numbers (CN) of 5–8, completing bonding distances of approximately 2.4 Å with coordination spheres [41]. In MalS, the relocation of Ca1 was coordinated by H464, N314, D406, and L397. Specifically, Ca1 was coordinated by the main-chain oxygens of H464 and L397 as well as the side-chain oxygens of N314 and D406, with coordinating distances of 2–2.4 Å (Figure 4A). Additionally, Ca1 was coordinated with one water molecule, which had high electron density. Ca2 was coordinated by N201, D203, N206, D207, D227, and G225. Specifically, Ca2 was coordinated by the main-chain oxygens of D203 and G225 as well as the side-chain oxygens of N201, N206, D207, and D227, with coordinating distances of 2.1–2.5 Å. Ca2 also had a water molecule participating in Ca coordination, leading to a high electron density. Therefore, Ca2 was likely coordinated by six residues and one water molecule (Figure 4B).

MalS has an optimal temperature of 42 °C for its activity and completely loses activity at 61 °C. At 55 °C, the activity of MalS is quenched. It is known that the binding of Ca^2+^ ions to MalS contributes to its thermostability [24]. Crystal structure analysis of AmyP_ΔSBD_ suggested that the stabilizing effect of bound Ca^2+^ was mainly due to Ca located adjacent to the -3 subsite. Ca promotes localized stabilization of the -3 subsite loop, which helps to preserve the enzyme’s catalytic activity [42]. In MalS, Ca1 was located around the -3, -4 subunit binding loop, and thus it stabilized the loop formed by residues 397–406 and participated in substrate binding at the active site (Figure 4E).

### 2.4. Structural Relevance of Maltohexaose-Releasing Activity of MalS

The catalytic site of the substrate in MalS is located in catalytic domain A, which adopts a (β/α)_8_ fold. Multiple sequence alignments revealed highly conserved catalytic residues, including D460, E503, and D565, suggesting a common catalytic strategy (Appendix A). Similar to the catalytic action of common amylases, D460 acts as a nucleophile, E503 as a proton donor, and D565 as a transition state stabilizer. MalS recognizes the substrate at the non-reducing end and hydrolyzes the 1,4-glucosidic bond. In a previous study, maltohexaose was reported as the major product formed by MalS from amylose at the earlier reaction stage [9]. In this study, we confirmed that when amylopectin was used as a substrate for MalS, the molar ratio of maltohexaose was the highest among small maltodextrin products at the initial reaction stage (Figure 5). Additionally, maltopentaose was formed in abundance, suggesting that MalS preferentially releases longer products first. Notably, maltodextrin with a degree of polymerization greater than 6 glucose units was not detected in the reaction.

To understand the structural rationale for this catalytic property of MalS, we compared the structures of MalS and CGTase (*Niallia circulans*, cyclomaltodextrin glucanotransferase, 1eo5). In this study, we used the Coot and CB-Dock2 programs to perform docking using the CGTase substrate as a model. By docking the substrate to MalS, we confirmed that MalS has a binding pocket capable of containing 6 glucose units. Additionally, the substrate binding pocket of MalS is negatively charged (Figure 6A).

In MalS, residue Y275 binds to subsite -1 of the substrate, and its side chain forms a π-π interaction that is common to the GH13 family. Structural comparison with CGTase (1eo5) revealed that other residues, including Y275, Y276, Y318, D385, A402, F403, R458, L607, D605, T263, F367, and R611, also contributed to substrate binding (Figure 6B). CGTase releases seven glucose units, with residue D147 present at the terminal position of the 7th glucose (-7 glucose) [43]. Comparing the structures of MalS and CGTase, we found that MalS has six glucose-tolerant binding pockets, with D385 present at the -6th glucose terminal (D147 in 1ea5). Additionally, F367 is located at the end of the -6th glucose unit and is expected to be involved in determining the number of glucosyl units in MalS (Figure 6C). Consequently, residues D385 and F367 are expected to play important roles in the preference of MalS for maltohexaose release and the hydrolysis of large dextrins.

MalS is also capable of digesting cyclodextrin, but much less efficiently than the linear substrate. As the embedded structure in previous studies, cyclodextrin mainly forms bonds with hydrophobic amino acids, particularly aromatic residues such as tyrosine, tryptophan, and phenylalanine [44,45]. In 1988, Freundlieb’s group regarded β-CD as a substrate of MalS and confirmed that MalS degraded β-CD [1]. In this study, we used the CB dock2 program to perform docking between MalS and β-CD. The residues predicted to be involved in MalS binding with β-CD were A402 and F403. However, the β-CD bound to F403 was far from the active site, which was a poor binding position. In contrast, residue A402 was near the catalytic position and therefore more likely than F403 to be the binding residue (Figure 6D). In MalS, cyclodextrin bound more weakly than the linear substrate, which was thought to be due to the weaker binding of residue A402 and the poor binding position of residue F403. Through our experiments, we found that MalS showed strong reactivity towards glycogen and amylopectin, whereas its reactivity towards β-CD and γ-CD was relatively weak. On the other hand, MalS had no effect on α-CD and pullulan (Figure 7). Moreover, we confirmed that MalS exhibited a stronger reverse effect on linear dextrin than on cyclodextrin. Figure 7 supports our conclusion that MalS exhibits low activity due to the weaker binding of residue A402 and the poor binding position of residue F403.

## 3. Materials and Methods

### 3.1. Materials

Amylopectin, glycogen, and pullulan were purchased from Sigma-Aldrich (St. Louis, MO, USA). The cyclodextrins (CD), including γ-CD, β-CD, and α-CD, were provided by Junsei Chemical (Tokyo, Japan).

### 3.2. Construction and Purification of Recombinant Protein

The genomic DNA of *E. coli* K12 was isolated using a genomic DNA extraction kit (G-spin^TM^ Genomic DNA Extraction Kit, Intron, Seongnam-si, Korea) and used for cloning the MalS gene. Oligonucleotides used for gene amplification were synthesized by GENOTEC Co. (Daejeon, Korea). The MalS gene, including its signal peptide, was amplified with the oligonucleotides 5′-ACTCATCCCATATGAAACTCGCCGCCTGTTTTC-3′ and 5′-AGCCG GAACTCGAGCTGTTGCCCTGCC-3′. The forward primer included a NdeI site, and the reverse primer included an Xho I site. The PCR product and the vector were approximately 2.4 kbp and 2.3 kbp in size, respectively. The recombinant plasmid (pTKNd6xH_MalS) was constructed to be 4.8 kbp in size to obtain the MalS protein. *E. coli* MC1061 was used as the cloning and protein expression host. *E. coli* MC1061 cells carrying the recombinant MalS were cultured overnight at 30 °C in Luria-Broth containing 50 µg/mL kanamycin. The target protein was purified in three steps. First, fast protein liquid chromatography was performed using nickel nitrilotriacetic acid resin (GH Healthcare, Chicago, IL, USA), then affinity chromatography was performed with amylose resin (New England Biolabs, Ipswich, MA, USA), and finally size exclusion chromatography was performed with HiLoad Superdex 200 pg (GE Healthcare, Chicago, IL, USA). The purity and activity of MalS were determined by the methods of Laemmli and Miller, respectively [46,47].

### 3.3. Crystallization and Data Collection

To initiate crystallization, 1 μL of MalS solution was mixed with 1 μL of crystallization solution containing 0.1 M sodium citrate tribasic dihydrate (pH 5.5), 20% *w*/*v* polyethylene glycol 1000, and 0.1 M lithium sulfate monohydrate. Protein crystals were obtained using the sitting-drop vapor diffusion method at 18 °C. Before data collection, the crystals were cryocooled at 100 K using a cryoprotectant solution consisting of 25% glycerol and mother liquor. Diffraction data were collected at a wavelength of 1 Å using beamline 7A at the Pohang Accelerator Laboratory (Pohang, Korea).

### 3.4. Structure Determination and Refinement

The collected diffraction data were processed using HKL2000 [48]. The molecular replacement method was used to determine the structure, employing the model provided by MRage, while the solvent content and Matthews coefficient were confirmed using phenix.xtriage of PHENIX [49,50]. Structural refinement and model building were carried out using PHENIX and the COOT [51] program. The structure was visualized using PyMOL [52] software. Detailed statistics for data collection and refinement are listed in Table 1.

### 3.5. Protein Docking and Alignment

Protein and ligand docking were conducted using the COOT program and CB-Dock2 website (https://cadd.labshare.cn/cb-dock2/php/index.php, accessed on 20 March 2023). Protein sequence alignment was performed using the Clustal Omega (https://www.ebi.ac.uk/Tools/msa/clustalo/, accessed on 20 March 2023) and SnapGene programs. A PDB similarity search was conducted using the Dali server (http://ekhidna2.biocenter.helsinki.fi/dali/, accessed on 20 March 2023).

### 3.6. Hydrolysis of Amylopectin by MalS

For understanding hydrolysis by MalS, an enzyme reaction with amylopectin (0.065 U/mg each substrate) was carried out at 50 °C in 50 mM MOPS (pH 7.0) containing 5.0 mM CaCl_2_ and sampled using the time-lapse method. The products were determined using HPAEC, as previously described with slight modifications. After column equilibrium with 150 mM NaOH, the samples were eluted with a gradient of 600 mM NaOAc in 150 mM NaOH at 1 mL/min [53]. The amount of each product was measured using the standard curve for each degree of polymerization under identical conditions.

### 3.7. Determination of Binding Affinity for Amylopectin and Glycogen 

After Ni-NTA purification, the interaction of MalS with ligands was determined by amylose affinity chromatography with the maltose-binding protein. MalS was loaded on the amylose resin; the unbound protein molecules were separated by Tris-HCl buffer (pH 7.5) and then first eluted by Tris-HCl buffer containing 0.5% amylopectin or 0.5% glycogen and then by Tris-HCl containing 50 mM maltose. For the control, MalS was eluted with Tris-HCl containing 50 mM maltose. Each elution fraction was analyzed by SDS-PAGE. 

### 3.8. Determination of Relative Activity

The relative activity of MalS was also determined on amylopectin, glycogen, pullulan, γ-CD, β-CD, and α-CD for comparison. The enzyme was reacted with each substrate for 10 min at 50 °C in 50 mM MOPS (pH 7.0) containing 5.0 mM CaCl_2_. The highest activity of MalS was considered the maximal relative activity [46,47].

### 3.9. Prediction of MalS Structure Using Alphafold2

To prediction the structure of MalS (UniProt: P25718), we utilized AlphaFold2 (https://colab.research.google.com/github/sokrypton/ColabFold/blob/main/AlphaFold2.ipynb, accessed on 3 May 2023) [54].

## 4. Conclusions

In this study, we have reported the first crystal structure of periplasmic amylase MalS, which belongs to GH13 subfamily 19, and we offer new insight into the structure–evolution relationship of periplasmic α-amylases. This study reveals that MalS has completely permutated domain structures and a unique CBM69 at the N domain. In addition, the crystal structure confirmed that the enzyme has a substrate binding pocket suitable to accommodate six glucose units. This study extends our understanding of GH13 subfamily 19 enzymes in terms of their evolutionary pathway as well as their structure/function relationship. 

## Figures and Tables

**Figure 1 molecules-28-03972-f001:**
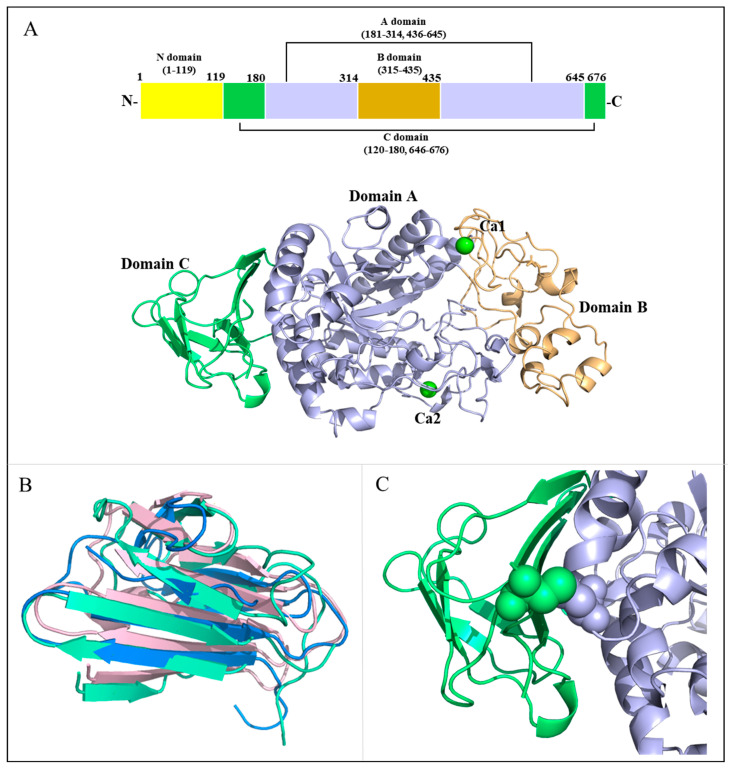
Overall structure of MalS. (**A**) Schematic overview of MalS with Ca^2+^-binding sites. The domains A, B, and C are shown in light blue, light orange, and cyan, respectively. The Ca^2+^ ions are shown in green. The disulfide bond between domains A and C is depicted as a sphere; (**B**) Structural alignment of domain C. The MalS protein neopullulanase 1 (1uh3) and cyclomaltodextrinase (1ea9) are shown in light pink and marine; (**C**) Disulfide-bridged site. The disulfide bond between domains A and C is depicted as a sphere.

**Figure 2 molecules-28-03972-f002:**
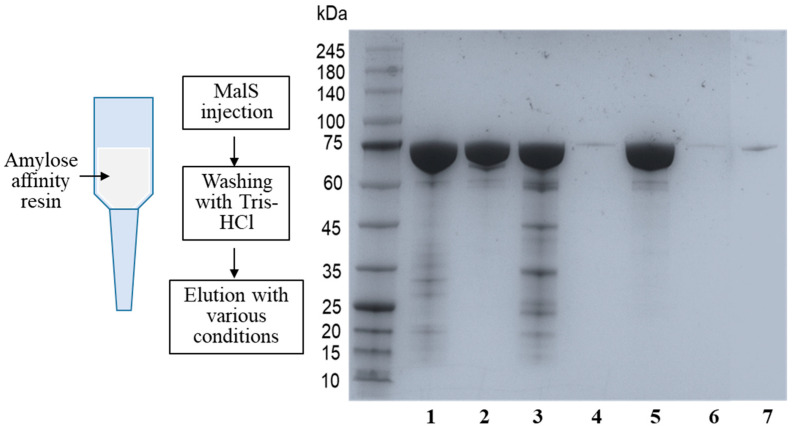
Affinity of MalS for amylopectin and glycogen. The affinity of MalS toward amylopectin and glycogen was evaluated using amylose affinity chromatography. MalS was subjected to amylose affinity chromatography and eluted under various conditions. Line 1: MalS before loading; Line 2: MalS after elution by Tris-HCl buffer containing 50 mM maltose; Lines 3 and 4: MalS elution using glycogen (Line 3) and then subsequent elution with 50 mM maltose (Line 4), respectively; Lines 5 and 6: MalS elution by amylopectin (Line 5) and then subsequent elution by 50 mM maltose (Line 6), respectively; and Line 7: MalS elution by Tris-HCl buffer.

**Figure 3 molecules-28-03972-f003:**
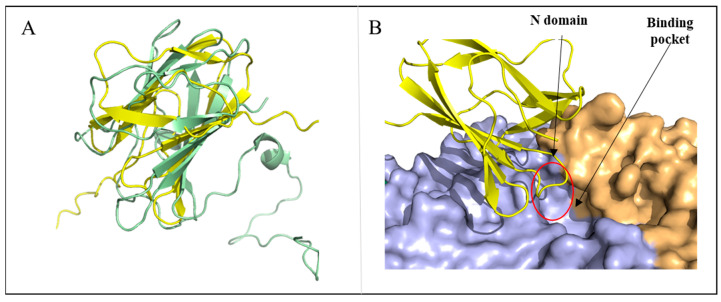
Structural characterization of N domain predicted by AlphaFold2. (**A**) Structure alignment of AlphaFold2-predicted N domain of MalS and CBM69 (5X5S, uncultured bacterium), with N domain shown in yellow and CBM69 shown in pale green. (**B**) The predicted structure of the N domain is located above the active site (red circle).

**Figure 4 molecules-28-03972-f004:**
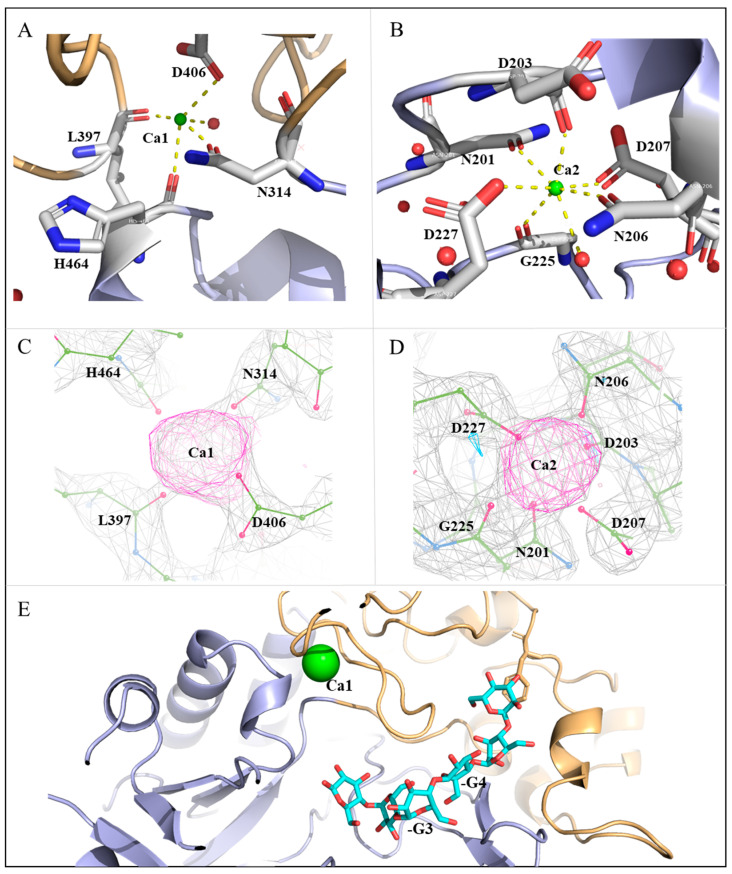
Calcium ion-binding sites in MalS. (**A**) Ca1 interacts with H464, N314, D406, L397, and water. The occupancy of Ca1 is 1.0, and the B-factor is 60.43 Å^2^; (**B**) Ca2 interacts with N201, D203, N206, D207, D227, G225, and water. The occupancy of Ca2 is 1.0, and the B-factor is 39.9 Å^2^. Ca^2+^ ions are depicted in green. Water molecules that interact directly with Ca^2+^ ions are shown in red; (**C**) The electron density omit map of *F*_o_ − *F*_c_ (3.35σ) shows the position of Ca1 and its binding residues; (**D**) The electron density omit map of *F*_o_ − *F*_c_ (3.35σ) shows the position of Ca2 and its binding residues; (**E**) The Ca1-surrounded loop is involved in substrate interactions (maltohexaose in cyan).

**Figure 5 molecules-28-03972-f005:**
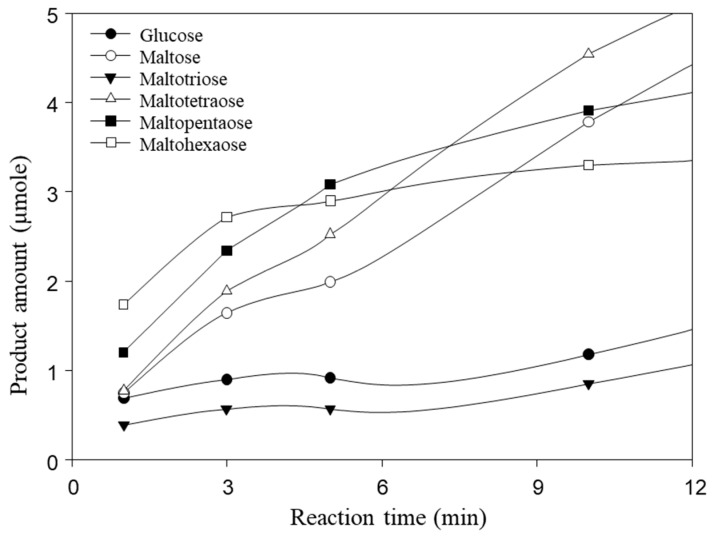
Hydrolysis products of MalS from amylopectin. Maltohexaose is produced as a main product at an early stage. The hydrolysis of 0.5% amylopectin with MalS was carried out at 50 °C in 50 mM MOPS (pH 7.0) containing 5.0 mM CaCl_2_. Maltohexaose was produced as the major product at the early stage.

**Figure 6 molecules-28-03972-f006:**
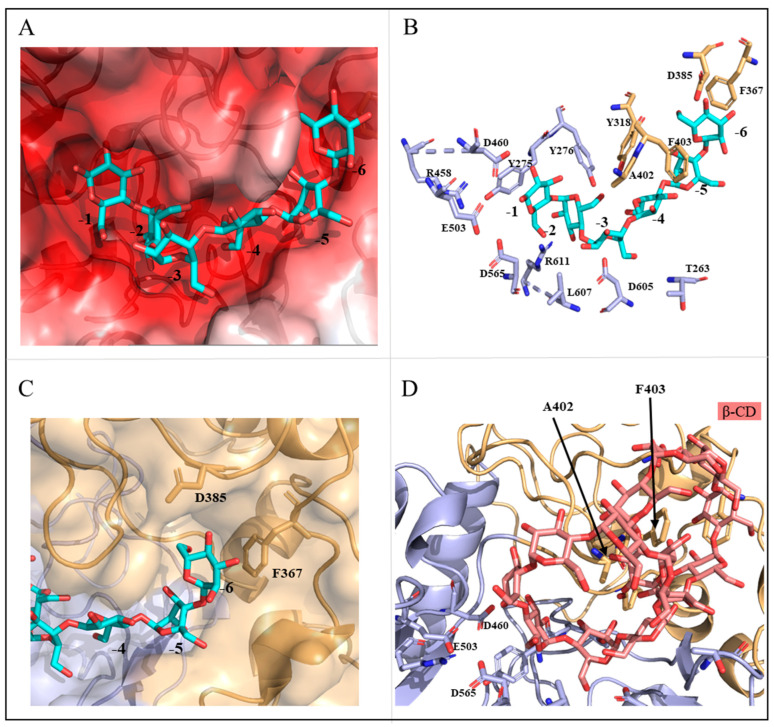
Structural analysis of catalytic residues in MalS. (**A**) Shows the charge distribution of the substrate binding pocket; (**B**) Displays the substrates and surrounding residues within the binding pocket of MalS; (**C**) Identifies the key residues involved in the binding of maltohexaose at the -6 subsite; (**D**) Depicts the possible binding sites of β-CD (salmon).

**Figure 7 molecules-28-03972-f007:**
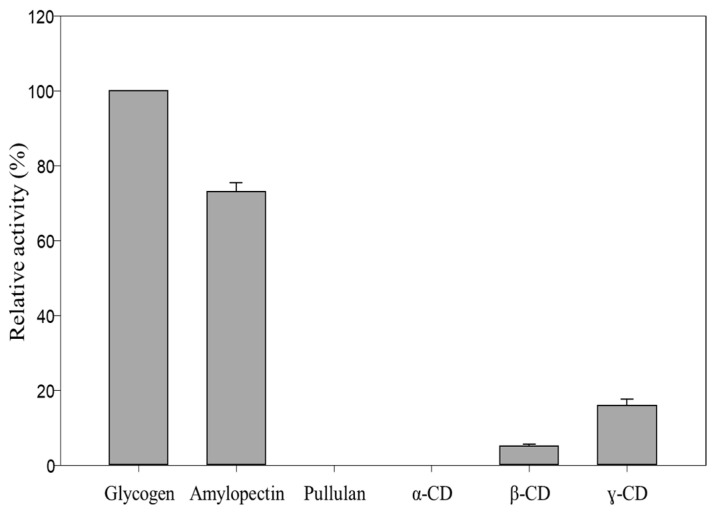
Relative activity of MalS toward glycogen, amylopectin, and α-, β-, and γ-CD.

**Table 1 molecules-28-03972-t001:** Data collection and processing statistics.

Data Statistics	
Space Group	R 3:H
Unit-cell parameters (Å, °)	A = 133.24, b = 133.24, c = 386.27 α = 90, β = 90, γ = 120
Unique reflections	69977
*R_sym_* (%) ^b^	12.3 (68.4)
Completeness (%)	99.8 (100) ^a^
Average *I/δ (I)*	168.0/5.5 (24.1/4.7)
Refinement statistics	
Resolution range (Å)	20.02-2.70
R factor (%) ^c^	0.178
*R*_free_ (%)	0.234
R.m.s.d., bond lengths (Å)	0.53
R.m.s.d., angle (°)	0.70
Average B value (Å^2^)	48
No. of atoms	18048
Ramachandran plot, residues in (%)	
Most favoured region	96%
Additionally allowed region	6%
Outlier region	0

^a^ Numbers in parentheses are statistics from the highest resolution shell. ^b^ Rsym =|I_obs_ − I_avg_|/I_obs_, where I_obs_ is the observed individual reflection and I_avg_ is the average over symmetry equivalents. ^c^ R factor = |F_0_| − |F_c_||/|F_0_|, where F_0_ and F_c_ are the observed and calculated structure factor amplitudes, respectively. R free was calculated using 5% of the data.

## Data Availability

The data presented in this study are available on request from the authors.

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
