# Peer review of "The Distinctive Permutated Domain Structure of Periplasmic α-Amylase (MalS) from Glycoside Hydrolase Family 13 Subfamily 19"

_molecules, 2023, doi:10.3390/molecules28103972_

Round 1

Reviewer 1 Report

An et al have presented a structural and biochemical characterization of the E. coli MalS protein. This is notable as this seems to be the first crystal structure of a member of this sub family within GH13 and the enzymes seems to have a unique domain organization with a circular permutation present, distinct from that seen in GH70 enzymes. The work is high quality and important, however, I think the authors are underselling the unique aspects a bit and so a bit more detail could be added.

Abstract

Italics for E. coli in the abstract

20 I believe this should be K12, not K21

Introduction

54 – Consistency should be maintained in how cyclodextrins are referred to, so beta- gamma- etc. If the number of glucose units is relevant, this can be included as extra information.

60 – when discussing the structure of GH13 enzymes it should be noted that the B-domain is inserted within the A domain.  

70 – do you mean that it is longer than alpha-amylases that do not contain other domains? Many microbial extracellular amylases contain a number of CBMs and are quite a bit larger than this, while intracellular ones without CBMs tend to be smaller. I’m not sure it is easy to say what the average amylase is like.

76-79 Maybe need to explain a bit more about what is unique about the N-terminus. It is very common for amylases to have a CBM48 at the N-terminus for instance.

Results and Discussion

93 so the C-domain is split between the N-terminus and the C-terminus, suggesting a circular permutation of the gene? It looks like this is true based on the presented structure and the alpha-fold prediction. How universal is this in the GH13_19 subfamily? To the best of my knowledge this has never been noted before. A figure showing this domain organization and an alignment with other members of the subfamily should be presented. Note that I don’t think this rearrangement is present in GH13_19 enzymes present in Gram positive bacteria, at least the two I checked. A comparison with the circular permutation present in most GH70 enzymes seems warranted as well.   

117 Please note that this is in subfamily GH13_19. This is highly relevant as this seems to be the first crystal structure solved for this subfamily, which seems critical given the apparent circular permutation of the gene presented.

143 please actually insert the reference for alpha fold

Figure 5 legend, please include some details about how the hydrolysis products were measured and conditions under which they were measured (e.g. temp and pH, concentration of substrate)

Materials and methods

Please include details of the production and purification of MalS

Information should be provided about the PDB entry for this new structure (I assume to be released upon publication)

272 please include the URL for CBDOCK2

278 either provide a reference or the exact details about how the BCA assay was conducted

Author Response

We highly appreciate the constructive comments from the reviewers. We believe it has enhanced the quality of the manuscript. We addressed all the comments of the reviewers and incorporated the necessary changes in modified version of the manuscript.

Abstract

Italics for E. coli in the abstract

Response:

It has been corrected. (line 21)

20 I believe this should be K12, not K21

Response:

It has been corrected. (line 21)

Introduction

54 Consistency should be maintained in how cyclodextrins are referred to, so beta- gamma- etc. If the number of glucose units is relevant, this can be included as extra information.

Response:

It has been changed to β-CD and γ-CD (line 57-58)

60 when discussing the structure of GH13 enzymes it should be noted that the B-domain is inserted within the A domain.

Response:

As suggested, we added the sentence “Domain B connects third α-helix and third β-sheet of domain A”. (line 72-73)

70 – do you mean that it is longer than alpha-amylases that do not contain other domains? Many microbial extracellular amylases contain a number of CBMs and are quite a bit larger than this, while intracellular ones without CBMs tend to be smaller. I’m not sure it is easy to say what the average amylase is like.

Response:

We are agreed with your concerns. We have modified that properly: MalS is composed of 676 amino acids, making it longer than typical α-amylases that are secreted extracellularly and have no N-terminal domains.

76-79 Maybe need to explain a bit more about what is unique about the N-terminus. It is very common for amylases to have a CBM48 at the N-terminus for instance.

Response:

Thank you for comments. We modified the sentences as below: MalS has more than 100 additional amino acids at the N-terminal. Although, previous studies of MalS demonstrated that the enzyme exhibits catalytic properties similar to typical extracellular α-amylases, this domain structure suggests that MalS has a distinct feature. (line 92-94)

Results and Discussion

93 so the C-domain is split between the N-terminus and the C-terminus, suggesting a circular permutation of the gene? It looks like this is true based on the presented structure and the alpha-fold prediction. How universal is this in the GH13_19 subfamily? To the best of my knowledge this has never been noted before. A figure showing this domain organization and an alignment with other members of the subfamily should be presented. Note that I don’t think this rearrangement is present in GH13_19 enzymes present in Gram positive bacteria, at least the two I checked. A comparison with the circular permutation present in most GH70 enzymes seems warranted as well.

Response:

The sentence was added in result part1: 

Intriguingly, domain C in MalSΔN is formed by the combination of N-terminal residues 120-180 and C-terminal residues 646-676. To our knowledge, this is the first C-domain structure in GH13 subfamily 19 that consists of two fragmented regions. Whereas, among GH70 enzyme, 4-α-glucanotransferases domains also have fragmented regions (Figure S4) [34]. Therefore, the evolutionary path of the MalS gene could be different from those of other GH13 subfamily 19 amylases. Sequence homology of MalS and those enzymes are very low. (line 117-122)

117 Please note that this is in subfamily GH13_19. This is highly relevant as this seems to be the first crystal structure solved for this subfamily, which seems critical given the apparent circular permutation of the gene presented.

Response:

Thank you for your comments. The GH13 subfamily, which includes MalS, was presented in the introduction (68-69). And please refer to the above answer. (line 60-69)

143 please actually insert the reference for alpha fold

Response:

It has been added. (line 163)

Figure 5 legend, please include some details about how the hydrolysis products were measured and conditions under which they were measured (e.g. temp and pH, concentration of substrate)

Response:

Thank you very much for the comment. The legend of Figure 5 has been edited as follows “Figure 5: Hydrolysis products of MalS from 0.5% amylopectin carried out at 50°C in 50 mM MOPS (pH 7.0) containing 5.0 mM CaCl2. Maltohexaose was produced as the main product at the early stage (line 231-233).

Materials and methods

Please include details of the production and purification of MalS

Response:

Thank you for reminding us of this content. The production and purification of MalS have been supplemented in the 3.2 section (line 281-288). Because of this supplementation, two more references are also added.

Information should be provided about the PDB entry for this new structure (I assume to be released upon publication)

Response:

We added the PDB ID(8IM8) in the overall structure part. (line 104)

272 please include the URL for CBDOCK2

Response:

It has been added. (line 306)

278 either provide a reference or the exact details about how the BCA assay was conducted

Response:

We sincerely apologize for the mistake in this method. For this result, we determined the amount of MalS products hydrolyzing from amylopectin based on the standard curve of each degree of polymerization analyzed by only high-performance anion exchange chromatography (HPAEC) performing at the same condition for sample analysis. Therefore, the method in this section has been edited with an emphasis on the HPAEC method (lines 310-316). Subsequently, its reference is added.

Reviewer 2 Report

I have no significant concerns. The work is scientifically valid, and the manuscript is well-written. A few modest comments.

Don't neglect to capitalise the name of the bacteria in the Abstract.

L43. Consider rewriting because it is unclear what was intended by "the transport of glucose units is limited."

L80. Why is the resolution so low?

L94. Why domain N is missing? 

L233. It is common for some amylase to degrade cyclodextrin. In your instance, can you give some quantitative value?

Author Response

We highly appreciate the constructive comments from the reviewer. We believe it has enhanced the quality of the manuscript. We addressed all the comments of the reviewers and incorporated the necessary changes in modified version of the manuscript.

L43. Consider rewriting because it is unclear what was intended by "the transport of glucose units is limited."

Response:

The sentence has been changed as follow.

MalF, MalG, and MalK form binding complexes to transport maltooligosaccharides across the cytoplasmic membrane. These complexes are responsible for transporting maltooligosaccharides with limited glucose units. (line 44-46)

L80. Why is the resolution so low?

Response:

The resolution of X-ray crystallography in the crystals of macromolecules is limited by the degree of crystal order, also known as "crystal quality". The resolution of the diffraction pattern depends on the degree of alignment of the crystals. When the crystal is highly ordered, the atoms are in defined positions throughout the crystal over time, and the crystal diffracts at high resolution. As the disorder increases, either from atoms moving over time or from the content of one unit cell differing from the next, the intensity of the diffraction spots decreases, and the resolution of the diffraction image is lower.

L94. Why domain N is missing?

Response:

From the AlphaFold structure provided by UniProt, we can notice that the N-domain is connected solely by a loop. As a result, the N-domain is expected to be highly flexible. Such flexible structures are difficult to position accurately in crystals and produce weak diffraction spots in X-rays. As a result, it is not possible to confirm the electron density map of the N-domain from diffraction data.

L233. It is common for some amylase to degrade cyclodextrin. In your instance, can you give some quantitative value?

Response:

Thank you very much for the comment. The figure 7 has been added. (line 273). The relative activities against amylopectin, glycogen, pullulan and cycloalodextrins was performed.

The sentence was added in result.(line 268-272)

Through our experiments, we found that MalS shows strong reactivity towards glycogen and amylopectin, whereas its reactivity towards β-CD and γ-CD is relatively weak. On the other hand, MalS has no effect on α-CD and pullulan (Figure 7). Moreover, we confirmed that MalS exhibits a stronger reverse effect on linear dextrin compared to cyclodextrin. Our conclusions are supported by this experiment.

Reviewer 3 Report

The manuscript entitled “Structural characterization analysis of periplasmic α-amylase (MalS), different from typical α-amylase, from Escherichia coli” reports on structure determination and a few characteristics of a periplasmic carbohydrate hydrolyzing enzyme of E. coli origin. The characterization seems to be for the full-length enzyme (MalS) and the structure determination for the truncated version (MalS∆N), lacking N-terminal domain. It is not clear how the authors obtained MalS or MalS∆N and what was the purpose of the study. Was the truncated enzyme cloned and produced recombinantly? If yes, details of cloning and recombinant production with the purpose of the study will strengthen the manuscript. Furthermore, a comparison of characteristics of MalS and MalS∆N should be added.

Minor comments:

1.    Line 2: Title should be modified by omitting “analysis”.

2.    Line 44: Cyclodexrins are cyclic compounds. Do they have reducing and non-reducing ends?

3.    Line 71: Reference for authors previous work is needed.

4.    Line 81: Elaborated.

5.    Line 88: E.coli

6.    Line 89: A picture of the crystals formed should be included.

7.    Line 92: Is not domain N missing in MalS∆N? What does “domain C (residues 120-180, 646-676)” mean? Why two different stretches of sequence?

8.    Numbering of the residues involved in coordination with calcium ions in Fig. 1 is wrongly labelled. Furthermore, these residue numbers do not match the numbers shown in Figure S1.

9.    Relative activities against amylopectin, glycogen, pullulan and cycloalodextrins should also be included.

10. Line 280: Details of determination by HPAEC should be given.

11.  Line 283: Any reason for performing Ni-NTA purification?

12. Line 299: No link is provided.

Language of the manuscript needs to be improved.

Author Response

We highly appreciate the constructive comments from the reviewer. We believe it has enhanced the quality of the manuscript. We addressed all the comments of the reviewers and incorporated the necessary changes in modified version of the manuscript.

1. Line 2: Title should be modified by omitting “analysis”.

Response:

It has been removed. (line 1)

2. Line 44: Cyclodexrins are cyclic compounds. Do they have reducing and non-reducing ends?

Response:

The sentence has been changed as follewed.

MalS degrades linear dextrins by recognizing the substrate at the non-reducing end and preferentially liberating maltohexaose. This product can then be transported across the cytoplasmic membrane via maltose ABC transporters. (line 46-49)

3. Line 71: Reference for authors previous work is needed.

Response:

We have added the UniProt accession numbers for each protein, which can be used to find their corresponding amino acid sequences. (line 86-90)

4. Line 81: Elaborated.

Response:

It has been corrected. (line 96)

5. Line 88: E.coli

Response:

It has been corrected. (line 103)

6. Line 89: A picture of the crystals formed should be included.

Response:

We apologize, but we do not have a crystallographic image of MalS as the data was collected a long time ago.

7. Line 92: Is not domain N missing in MalS∆N? What does “domain C (residues 120-180, 646-676)” mean? Why two different stretches of sequence?

Response:

We used the full-length protein to make crystals. However, Domain N was not experimentally observable in the structure of the MalS protein. Based on the AlphaFold structure provided by UniProt, it is evident that the N-domain is connected only by a loop, indicating high flexibility. Such flexible structures pose challenges in accurate positioning within crystals, often resulting in weak diffraction spots in X-ray analysis. Consequently, confirming the electron density map of the N-domain from diffraction data is not feasible. However, in this study, the N-domain was analyzed using the predicted structure from AlphaFold.

About the C domain, the sentence was added in result.

Intriguingly, domain C in MalSΔN is formed by the combination of N-terminal residues 120-180 and C-terminal residues 646-676. To our knowledge, this is the first C-domain structure in GH13 subfamily 19 that consists of two fragmented regions. Whereas, among GH70 enzyme, 4-α-glucanotransferases domains also have fragmented regions (Figure S4) [34]. Therefore, the evolutionary path of the MalS gene could be different from those of other GH13 subfamily 19 amylases. Sequence homology of MalS and those enzymes are very low. (line 117-122).

8. Numbering of the residues involved in coordination with calcium ions in Fig. 1 is wrongly labelled. Furthermore, these residue numbers do not match the numbers shown in Figure S1.

Response:

We have checked that Figure 1 and the Ca ion number is correct. Is the figure presented here correct for Figure S1? Figure S1 is not related to Ca coordination.

9. Relative activities against amylopectin, glycogen, pullulan and cycloalodextrins should also be included.

Response:

Based on the comment, the relative activity of MalS against amylopectin, glycogen, and cyclomaltodextrin has been supplemented. They were presented in Figure 7 with the legend below. (line 268-274)

10. Line 280: Details of determination by HPAEC should be given.

Response:

Thank you very much for the comment. We strongly agree that the HPAEC method should be more details. Therefore, it has been edited (line 310-316).

11. Line 283: Any reason for performing Ni-NTA purification?

Response:

In this study, MalS was constructed as the His-tagged protein. Therefore, Ni-NTA was used for the purification of MalS. By this purification, most of the non-target proteins were removed. However, to get the high purity of MalS for structure analysis, two more purification steps, including affinity chromatography and size exclusion chromatography, were carried out.

12. Line 299: No link is provided.

Response:

We will add the Supplementray data linker.

Round 2

Reviewer 1 Report

Manuscript is in good shape. My one additional comment is that a bit more could be done to compare to other GH13_19 subfamily members. The authors have included the line (127-129 in the marked revised manuscript): 

"Therefore, the evolutionary path of the MalS gene could be different from those of other GH13 subfamily 19 amylases. Sequence homology of MalS and those enzymes are very low."

Enzymes from the same sub-family tend to have relatively high overall sequence homology, so I am not sure what exactly is meant by this line. Also, if I look in the CAZy database there are 8 members of GH13_19 that are classified as characterized and have Uniprot entries (including MalS). Quickly looking at those in Alphafold shows that the Klebsiella enzyme also has this split C-domain, showing it is not unique to MalS, however, the other members (which are Gram Positives) do not have this. So, it would appear that this may be restricted to Enterobacteriaceae GH13_19 enzymes. Of course a more detailed bioinformatic analysis may be required to fully define the distribution which may be beyond the scope of this paper, but I think a supplementary alignment of those characterized members of the subfamily may be warranted or at least a rewording of lines 127-129. 

Author Response

We greatly appreciate the constructive comments provided by the reviewer. We have carefully addressed the comments raised by the reviewer and have incorporated the necessary changes in the revised version of the manuscript. The revised sections are also highlighted in red for easy reference.

Response: Thank you for comments. We modified the sentences as below (line 146-167):

Intriguingly, domain C in MalSΔN is formed by the combination of N-terminal residues 120-180 and C-terminal residues 646-676. To our knowledge, this is the first C-domain structure in GH13 subfamily 19 that consists of two fragmented regions. Whereas, among GH70 enzyme, GTF180-ΔN (glucansucrase, Lactobacillus reuteri 180, PDB: 3KLK, GH70) folds into five distinct linearly arranged domains, designated as A, B, C, IV, and V. The structure of GTF180-ΔN reveals that, contrary to the linear arrangement, four of the five domains are organized in discontinuous parts of the polypeptide chain. Only the C domain is formed from one continuous stretch of amino acids (Figure S4A). Additionally, the N-terminal variable domain, which is not present in the crystal structure, is not required for correct folding of the enzyme. This structure also confirms that the catalytic (β/α)8-barrel domain of GTF180-ΔN undergoes circular permutations, as previously suggested. Among the MalSΔN, it has been observed that the C domain is organized in discontinuous parts of the polypeptide chain, suggesting the presence of circular permutations in this domain (Figure S4B). The "permutation per duplication model" proposes a series of gene arrangements that could have resulted in a circularly permuted gene through gene duplication, in-frame fusion, and partial truncation processes.

MalS is highly similar in sequence and Alphafold-predicted structure to Maltohexaose-producing amylase from Klebsiella pneumoniae (Uniprot: Q9RHR1), both of which belong to the same GH13_19 family, and share C domains composed of discontinuous parts of the polypeptide chain. Moreover, even when referring to periplasmic alpha-amylases from different species, such as Shigella flexneri (Uniprot: A0A0H2V321), Salmonella typhimurium strain LT2 / SGSC1412 / ATCC 700720 (Uniprot: Q8ZL87), Citrobacter rodentium strain ICC168 (Uniprot: D2TJT7), etc., the sequences and Alphafold-predicted structures are highly similar. This suggests that MalS likely originated from a common ancestor and has been conserved over a long evolutionary period. In other words, since MalS plays an important role in the survival of various bacteria, its conservation across diverse bacterial types further supports its ancient origin and evolutionary significance.

Four more references are also added.

Reviewer 3 Report

The authors did not respond to the major points including how they obtained MalS or MalS∆N and what was the purpose of the study. Was the truncated enzyme cloned and produced recombinantly? Have they determined the N-terminal amino sequence? If yes, details of cloning and recombinant production with the purpose of the study will strengthen the manuscript.

Minor points:

1)    Fig. 4B: Residue numbers are wrongly labeled It cannot be D227 and G227.

2)    Line 406: Data Availability Statement.

Language of the manuscript needs attention.

Author Response

We greatly appreciate the constructive comments provided by the reviewer. We have carefully addressed the comments raised by the reviewer and have incorporated the necessary changes in the revised version of the manuscript. The revised sections are also highlighted in red for easy reference.

Comments and Suggestions for Authors

The authors did not respond to the major points including how they obtained MalS or MalS∆N and what was the purpose of the study. Was the truncated enzyme cloned and produced recombinantly? Have they determined the N-terminal amino sequence? If yes, details of cloning and recombinant production with the purpose of the study will strengthen the manuscript.

Response:

Thank you for comments. We changed sentences as below:

Title:

The distinctive permutated domain structure of periplasmic α-amylase (MalS) from glycoside hydrolase family 13 subfamily 19

Abstract:

(line 22-28)

A periplasmic α-amylase, MalS (EC. 3.2.1.1) that belongs to glycoside hydrolase (GH) family 13 subfamily 19, of Escherichia coli K12, is an integral component of the maltose utilization pathway and used among enterobacteriaceae for the effective utilization of maltodextrin. We present crystal structures of MalSΔN from E. coli and reveal that it has unique structural features of circularly permutated domains and a possible CBM69. The C-domain of MalSΔN consists of amino acids 120-180 and 646-676, and the whole domain architecture shows the complete circular permutation of C-A-B-A-C in domain order.

(line 37-39)

Structural analysis of MalS provides new insights into the structure-evolution relations in GH13_subfamily 19 enzymes and a molecular basis for understanding the details of catalytic function and substrate binding of MalS.

Conclusions

(line 397-403)

In this study, we have reported the first crystal structures of periplasmic amylase MalS that belong to the GH13_subfamily 19, and offer new insights into the structure-evolution relations of periplasmic α-amylases. This study reveals that MalSΔN has completely permutated domain structures and a unique CBM69 at N-domain. In addition, the crystal structure approved that the enzyme has a substrate binding pocket suitable to accommodate six glucose units. This study extended our understanding of the GH 13_subfamily 19 enzymes in terms of their evolutionary pathway as well as structure/function relations.

We utilized the full-length MalS protein without undergoing enzymatic cleavage as a recombinant in our study. The crystals were obtained using the full-length protein. However, upon analyzing the crystal structure data, we were unable to confirm the presence of the N domain. This is likely due to the high flexibility of N domain.

We added sentence as below (line 334-343):

The genomic DNA of E. coli K12 was isolated using a genomic DNA extraction kit (G-spinTMGenomic DNA Extraction Kit, Intron, Seongnam-si, Korea) and used for cloning the MalS gene. Oligonucleotides used for gene amplification were synthesized by GENOTEC Co. (Daejeon, Korea). The MalS gene, including its signal peptide, was amplified with the oligonucleotides 5’-ACTCATCCCATATGAAACTCGCCGCCTGTTTTC–3’ and 5’-AGCCG GAACTCGAGCTGTTGCCCTGCC-3’. The forward primer included a Ndeâ…  site, and the reverse primer included an Xhoâ…  site. The PCR product and the vector were approximately 2.4 kbp and 2.3 kbp in size, respectively. The recombinant plasmid (pTKNd6xH_MalS) was constructed to be 4.8 kbp in size to obtain the MalS protein. E. coli MC1061 was used as the cloning and protein expression host.

Minor points:

1) Fig. 4B: Residue numbers are wrongly labeled It cannot be D227 and G227.

Response: It has been corrected. (line 251)

2) Line 406: Data Availability Statement.

Response: It has been corrected. (line 422)